# The problem is escalating: Barriers faced by medical students in conducting research; A scoping literature review

**L. Winter Mokhwelepa**[1]*, **G. Olivia Sumbane**[1], **Shisana Baloyi**[2], **Samuel Risenga**[3], **Tumisang Malete**[4]

**1** School of Medicine, University of Limpopo, Polokwane, South Africa, **2** Department of Obstetrics & Gynecology, University of Limpopo, Polokwane, South Africa, **3** Department of Pediatrics, University of Limpopo, Polokwane, South Africa, **4** Department of internal medicine, University of Limpopo, Polokwane, South Africa

* 201826851@myturf.ul.ac.za

## Abstract

### Background

Research is a cornerstone of medical education, equipping future healthcare professionals with critical thinking skills and the ability to apply evidence-based practices. Despite its importance, medical students often encounter numerous obstacles that hinder their active participation in research. These challenges not only affect students' academic growth but also limit the advancement of medical knowledge and innovation within the healthcare system.

### Aim

This study aimed to review, map, and synthesize the barriers that hinder medical students from engaging in research, and to provide insights that can inform educational reforms and policy interventions to strengthen research capacity among medical students.

### Methods

The authors followed the Joanna Briggs Institute (JBI) Scoping review guidelines. The initial search was conducted in January 2025 and concluded in August 2025. A comprehensive literature search was performed across three major electronic databases: PubMed, Web of Science, and Scopus. The following key terms were used: "medical students", "research barriers", "research participation", "challenges", and "medical education". Only peer-reviewed studies published in English between 2010 and 2025 that examined barriers faced by medical students in conducting research were included. Studies focusing on non-medical or postgraduate students, non-English articles, editorials, and abstracts without full texts were excluded.

**Data availability statement:** No new data were created or analyzed in this study. Data sharing is not applicable to this research.

**Funding:** The author(s) received no specific funding for this work.

**Competing interests:** The authors have declared that no competing interests exists.

## Results

Fifty-seven studies met the inclusion criteria. The review identified key barriers hindering medical students' engagement in research, including limited mentorship, heavy academic workload, and insufficient research training. Institutional challenges such as inadequate funding and poorly structured curricula further restricted participation.

## Conclusion

Improving mentorship, research training, and institutional support is crucial to empower medical students and foster greater engagement in research.

## Introduction

Medical research is a cornerstone of medical education, fostering critical thinking and evidence-based practice. However, despite its significance, medical students encounter numerous barriers that impede their active participation in research activities. These obstacles not only hinder their academic and professional development but also limit the advancement of medical knowledge.

A predominant barrier is the lack of time. Medical curricula are often densely packed with clinical and theoretical coursework, leaving students with limited opportunities to engage in research. A study by Haran et al. (2023) found that 65% of medical students reported time limitations as a significant impediment to research involvement [1]. Similarly, Alhabib et al. (2023) identified time constraints, lack of mentoring, and lack of interest in research as the top three barriers influencing research conduct among medical students [2].

Financial constraints also play a critical role. Many medical students lack access to research funding or grants, which are essential for conducting meaningful research projects. A study by Kumar et al. (2019) revealed that 83.26% of participants considered lack of finance or funding to be a significant barrier to research [3]. This financial strain often leads to a reliance on external funding sources, which may not always align with the students' research interests or ethical standards. Additionally, inadequate mentorship and guidance are prevalent challenges. Many students report difficulty in finding experienced researchers willing to supervise their projects. The study by Orebi et al. (2023) highlighted that lack of mentorship was a significant barrier to research participation among medical students [4]. Without proper guidance, students may struggle with research design, data analysis, and interpretation, leading to suboptimal research outcomes [5].

Furthermore, limited access to research resources, such as databases and journals, hampers students' ability to conduct comprehensive literature reviews and stay updated with current research trends. Alsaleem et al. (2021) noted that limited access to medical journals and related databases were significant barriers to research activities among medical students [6]. This lack of access restricts students' ability to build upon existing knowledge and contribute novel insights to the field. The significance of

these barriers is emphasized by their impact on students' attitudes toward research [7]. A study by Quintero et al. (2025) found that academic overload, lack of mentorship, and insufficient training were frequently cited as major obstacles that limit student involvement in research [8]. These challenges not only discourage students from pursuing research but also affect the overall quality and quantity of medical research output [9].

A scoping review is particularly warranted at this point in time due to the rapid expansion of medical education programs globally and the growing emphasis on early research involvement as a core competency for future physicians. Despite this shift, evidence on barriers to medical student research engagement remains highly fragmented, with studies dispersed across regions, income settings, and educational systems, often addressing isolated factors without integrated synthesis. Existing narrative reviews tend to be descriptive and selective, while systematic reviews have focused narrowly on specific interventions or outcomes, limiting their ability to capture the full scope and contextual variability of reported barriers. This scoping review advances understanding by systematically mapping the breadth, types, and distribution of barriers across diverse geographic and institutional contexts, identifying underexplored regions, and highlighting gaps in policy and curricular responses. By providing a comprehensive, policy-relevant overview rather than evaluating intervention effectiveness alone, this review offers an evidence base to inform curriculum reform, institutional research capacity-building, and future targeted systematic reviews. Therefore, this study aimed to review the barriers that hinder medical students from engaging in research.

## Methods

The authors of this study conducted the scoping review in accordance with the Joanna Briggs Institute (JBI) [10] and reported according to the Preferred Reporting Items for Systematic Review and Meta-Analyses extension for Scoping Reviews guidelines (PRISMA) [11] (S1 Appendix). This scoping review approach aimed to synthesize the existing literature on barriers faced by medical students in conducting research.

### Ethics statement

This review did not involve primary data collection and therefore did not require ethical approval.

### Research question

*What are the barriers and challenges faced by undergraduate and graduate medical students in conducting research, and how do these barriers influence their research participation?*

### Inclusion criteria

Articles were included if they:

- Focused on undergraduate or graduate medical students.

- Discussed barriers or challenges in conducting research.

- Were published in English in peer-reviewed journals.

### Exclusion criteria

Articles were excluded if they:

- Focused solely on faculty or non-medical students.

- Were conference abstracts without full texts, letters, or non-peer-reviewed sources.

- Were published prior 2010

- Were not written in English language

## Search strategy

A structured literature search was conducted across three electronic databases, including PubMed, Scopus, and Web of Science. The search was initiated in January 2025 and concluded in August 2025. The university librarian was invited to assist with the literature search. The studies searched were written in English language. The restriction to English-language publications published between 2010 and 2025 was applied to ensure the inclusion of contemporary evidence that reflects current medical education structures, research training models, and institutional expectations. However, this decision may have shaped the scope of the findings by excluding relevant studies published in other languages, particularly from non–English-speaking regions, potentially limiting the global representation of reported barriers.

Search terms were developed based on the research question and included combinations of keywords and Boolean operators: "medical students" AND "research barriers", "medical education" AND "research participation challenges", "undergraduate medical research" AND "obstacles", "mentorship" OR "funding" OR "time constraints" AND "medical student research". Boolean operators (AND/OR) were systematically applied to broaden or narrow the search as appropriate, and truncation symbols (e.g., research, student) were used to capture variations in word endings and spelling across databases. Database-specific adaptations were implemented by modifying controlled vocabulary terms, such as MeSH headings in PubMed and subject headings in Scopus and Web of Science, to enhance search sensitivity and specificity. Search strings were iteratively refined in consultation with a medical librarian to ensure comprehensive retrieval of relevant studies. Initially, the review aimed to identify studies published within the last ten years (2015–2025) to capture recent trends in medical education and research engagement. However, during preliminary search, it became evident that the number of studies focusing specifically on barriers to research among medical students was limited. To ensure sufficient coverage and a more comprehensive analysis, the timeframe was extended to 2010–2025. This extension allowed the inclusion of earlier studies that still provide relevant insights into persistent and evolving barriers in medical student research participation. A full search strategy is shown in (S2 Appendix).

## Study selection

A total of 1,024 articles were initially retrieved. After removing 312 duplicates, 712 unique articles remained for screening. During the title and abstract screening, two independent reviewers assessed the articles for relevance based on predefined inclusion and exclusion criteria, focusing on studies addressing barriers to research participation among medical students. To enhance thoroughness, a manual screening of the reference lists of key articles was conducted to identify additional relevant studies that may have been missed by electronic searches. Furthermore, a medical librarian was consulted to refine search terms, ensure the inclusion of appropriate databases, and verify the comprehensiveness of the search strategy. Then, 128 proceedings were excluded because they didn't have full information to assist in this study. Following this process, 584 reports were sought for retrieval, of which 184 reports could not be retrieved, resulting in 400 reports assessed for eligibility. Upon full-text assessment, 343 reports were excluded for the following reasons: 150 studies were not available in English; 90 full-text articles could not be retrieved; 46 studies were published prior to 2010; 32 studies focused on non-medical students; 15 articles did not report barriers to research participation as a primary outcome; and 9 publications were non-peer-reviewed sources, including editorials, commentaries, and conference abstracts. Ultimately, 57 studies met all inclusion criteria and were included in the final synthesis for this scoping review. The summary of study selection is highlighted by Fig 1.

Screening was conducted by three independent reviewers to enhance methodological rigor. All reviewers first screened titles and abstracts against the predefined inclusion and exclusion criteria. Full-text articles were then assessed in duplicate by at least two reviewers independently to determine eligibility. Discrepancies or disagreements at any stage were resolved through discussion, and when consensus could not be reached, a third reviewer was consulted to make the final determination. This approach ensured consistent application of eligibility criteria and strengthened the reliability of categorizing studies according to reported barriers.

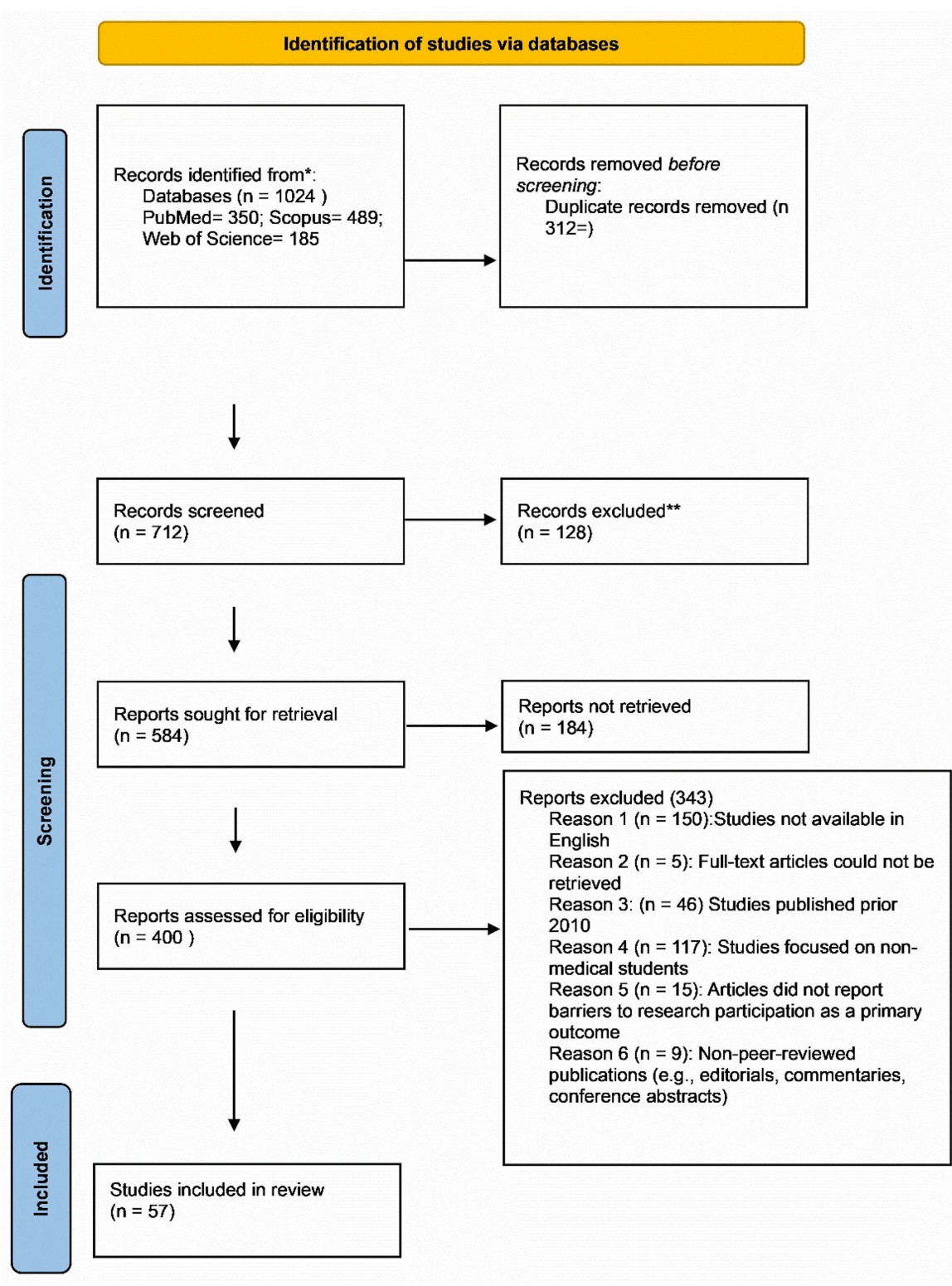

**Fig 1. Prisma flow diagram [11].**

## Data extraction and analysis

For this scoping review, a standardized data extraction form was developed to systematically capture relevant information from the included studies. Key details extracted included author(s) and year of publication, country of study, study design, population characteristics, and identified barriers to research. Data extraction summary is shown in (S3 Appendix). Three independent reviewers performed data extraction to ensure accuracy and minimize bias, with discrepancies resolved through discussion and consensus. Following extraction, the data were synthesized using thematic analysis, which involved identifying, analyzing, and reporting patterns or themes across the studies. This process included familiarization with the data, coding relevant information related to research barriers, clustering codes into overarching themes (e.g., time constraints, lack of mentorship, financial challenges, institutional support), and refining the themes to ensure coherence and clarity.

The identification of key barriers was conducted using a thematic synthesis approach, combining both inductive and deductive methods. Initially, barriers were extracted directly from each study and coded inductively to capture themes emerging from the data. These codes were then mapped onto pre-existing categories of common research barriers reported in the literature, allowing for a deductive refinement of themes. To provide insight into the relative prominence of barriers, the frequency of each barrier across included studies was tabulated, highlighting which challenges were most consistently reported. This mixed approach enabled a structured yet flexible synthesis, ensuring that both expected and novel barriers were captured, and facilitated a comprehensive understanding of the patterns and prevalence of obstacles faced by medical students engaging with research.

## Results

This review included fifty-seven studies that met inclusion criteria. Four themes emerged from this study: time constraints, lack of research mentorship, inadequate training and resources, and institutional barriers. This study revealed that most of the studies were conducted in the following countries respectively: Pakistan (n = 9), India (n = 7), Bangladesh (n = 3), Egypt (n = 3), United Arab Emirates (n = 2), Iran (n = 2), Nigeria (n = 2), Jordan (n = 2), New York USA (n = 1), Ecuador (n = 1), Spain (n = 1), Taiwan (n = 1), Romania (n = 1), Iraq (n = 1), Qatar (n = 1), Kuwait (n = 2), Oman (n = 1), Malaysia (n = 1), Sydney Australia (n = 1), Bahrain (n = 1), Turkey (n = 1), Syria (n = 1), Britain (n = 1), Sudan (n = 1), Morocco (n = 1) Brazil (n = 1), England (n = 1), Canada (n = 1), and Palestine (n = 1).

Furthermore, this study revealed that most studies included were of cross-sectional designs (n = 49), mixed method studies (n = 4), and qualitative studies (n = 5) respectively as illustrated in Fig 2. This suggests that the existing research on barriers faced by medical students in conducting research is mostly descriptive and quantitative, with limited in-depth qualitative exploration of students' personal experiences or contextual factors. Notably, all the studies reviewed revealed that barriers faced by medical students in conducting research are time constraints, lack of research mentors, inadequate training and resources, and institutional barriers.

## Theme 1: time constraints

A major finding across several studies was that medical students face significant time-related challenges when attempting to engage in research [1–9]. Five studies reported that medical students struggled to balance academic requirements, clinical rotations, and personal commitments, leaving little room for research activities [12–15]. In most cases, over two-thirds of participants indicated that time limitations were the greatest obstacle to research participation.

Similar studies emphasized that the intense structure of medical curricula prevents students from dedicating sufficient hours to research projects, even when they express genuine interest [6,16,17,19–30]. findings further affirmed that students perceive research as an "extra burden" rather than an integral part of their training, as academic and clinical pressures often take priority [31–40]. Collectively, these findings illustrate that without dedicated time allocations, student research participation is likely to remain low.

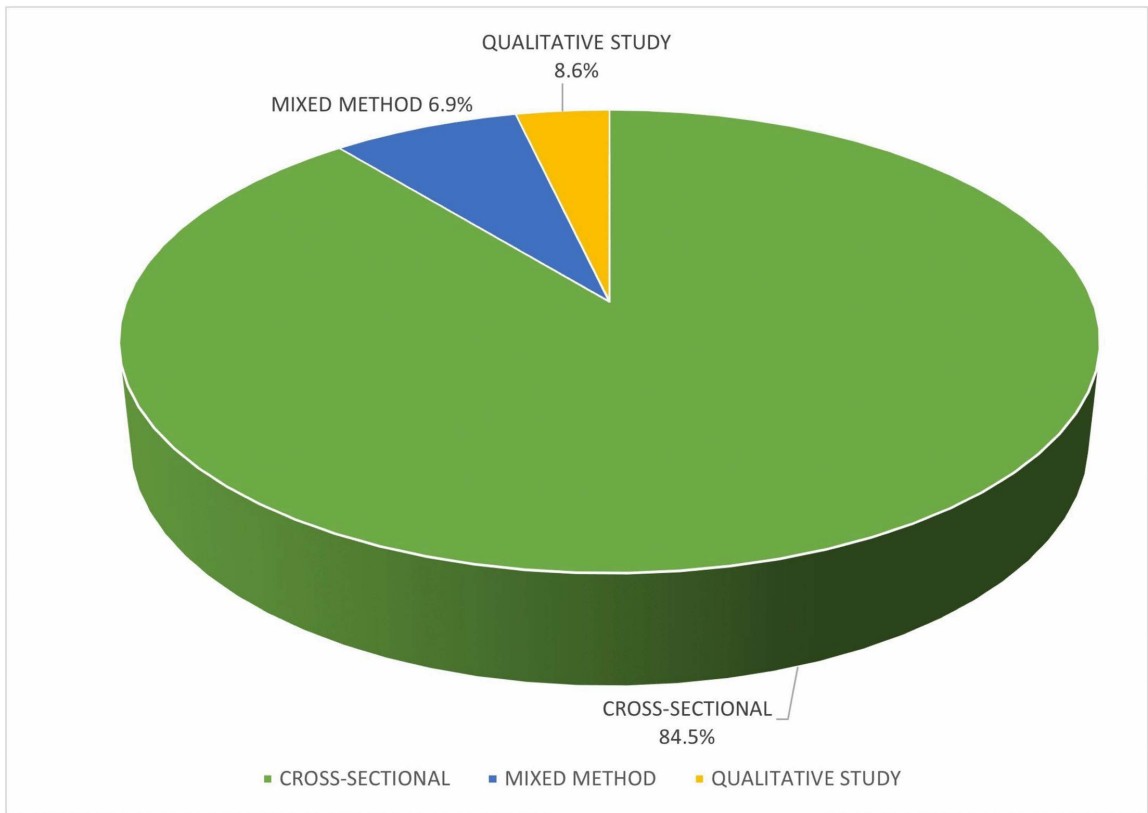

**Fig 2. The study designs included in the review.**

## Theme 2: lack of research mentorship

Another prominent finding from this review is the lack of effective mentorship. Six studies found that students commonly experienced difficulty finding supervisors to guide them through project design, data collection, and publication processes [41–50]. In several of these studies, more than 80% of participants identified inadequate supervision as a key barrier.

Similar research indicated that when mentorship is available, it is often inconsistent or limited to informal guidance [20–49]. Qualitative study affirmed that the absence of structured mentorship leads to feelings of uncertainty, lack of direction, and decreased motivation to pursue research [17]. This evidence highlights that mentorship is a foundational factor in promoting successful student research engagement.

## Theme 3: inadequate training and resources

Limited training and lack of access to resources emerged as another significant barrier. Four quantitative studies found that most students lacked basic skills in research methodology, statistical analysis, and academic writing, with many reporting minimal exposure to research education during undergraduate training [17–29]

Similar studies highlighted that insufficient access to materials, such as journals, laboratories, and funding, further discourages participation [16,17]. Qualitative evidence affirmed that the lack of structured workshops and statistical support units within institutions leaves students unable to translate theoretical knowledge into practical research skills [17,32,48]. These findings suggest that research education must be integrated early and consistently throughout medical curricula.

## Theme 4: Institutional barriers

Institutional-level challenges were frequently reported as systemic obstacles to student research participation [51–57]. Several cross-sectional studies found that rigid curriculum structures, lack of protected research time, and limited funding opportunities hindered engagement [49–56]. Other studies specifically identified that absence of institutional recognition for student research or lack of incentives reduced motivation to engage in scholarly activity [1–9, 12–50]

Interestingly, other studies further affirmed that poor administrative support, delays in ethics approval, and limited access to supervisors within universities contribute to students' perception that research is undervalued [39–45,47,50]. Together, these findings emphasize that institutional culture and infrastructure play a decisive role in shaping students' research engagement and productivity.

## Discussion

The barriers identified in this review: time constraints, lack of mentorship, inadequate training and resources, and institutional hurdles are reflective of systemic challenges that significantly impact medical students' engagement in research. However, it is important to differentiate these findings from mere descriptions of obstacles and delve deeper into why medical students experience these barriers more acutely than the general student population.

Firstly, the time constraints faced by medical students are inherently tied to the unique structure of medical education. Unlike students in many other disciplines who have relatively flexible academic schedules, medical students must navigate a demanding curriculum that combines extensive theoretical coursework with clinical responsibilities, often requiring long hours in hospital settings [58]. This dual role limits the availability of continuous and protected time for research activities, a challenge less commonly seen among other undergraduates or graduate students who do not have clinical duties [12–57]. This intense schedule not only restricts research time but also impacts students' mental bandwidth to engage deeply with research projects.

Secondly, the lack of effective mentorship in medical research is exacerbated by the competing demands placed on clinical faculty [1–9]. Medical educators often balance patient care, teaching, and administrative responsibilities, which limits their capacity to provide consistent research mentorship [59]. In comparison, students in purely academic or scientific disciplines may have easier access to mentors focused primarily on research. This scarcity of mentorship impedes the development of research skills and diminishes motivation among medical students, highlighting a structural issue that is less pronounced in other fields.

Moreover, the inadequate training and resources available to medical students illustrate a disconnect between the research skills taught and the practical application within clinical environments. Many medical curricula emphasize rote learning and clinical knowledge, with less integrated training in research methodologies and limited access to research infrastructure such as laboratories or data analysis [60]. This gap contrasts with research-oriented programs where students routinely engage in hands-on projects, thereby reinforcing a divide between medical students and their peers in research-rich disciplines.

Institutional barriers in medical schools are also distinctive because of the prioritization of clinical education and patient care, which often supersedes research facilitation for students [61–63]. Unlike other academic departments where research may be a core mission, medical institutions may lack streamlined processes or incentives to support student research initiatives, reflecting broader organizational priorities that shape the research landscape for medical students.

Overall, medical students' challenges in research appear to stem from the demanding, dual-focus nature of their training, limited availability of mentorship, and institutional priorities that often emphasize clinical competence over research engagement. These factors create a uniquely constrained environment compared with other student populations, as identified across the included studies. While the evidence highlights these barriers, the implications for institutional reform remain interpretive. Potential strategies suggested by literature and informed by the authors' perspective include embedding research training within the curriculum, strengthening mentorship programs, and providing dedicated resources to

support research engagement. Such measures could help integrate research more fully into medical training, though their feasibility and effectiveness require further evaluation in context-specific settings.

In support of integrating research training early within medical curricula, emerging evidence highlights the effectiveness of low-resource pedagogical interventions in addressing inadequate research skills. For example, a recent study by Rehman et al., 2024, demonstrated that structured peer feedback on research writing significantly improved students' academic writing quality, critical appraisal skills, and confidence in conducting research [63]. This approach directly addresses the training gaps identified in this review and offers a practical, scalable strategy that medical schools particularly in resource-constrained settings can adopt to strengthen research skill acquisition and student engagement in scholarly activities.

### Study recommendations

- Integrate Research Training Early: Incorporate comprehensive research methodology and practical skills training into the medical curriculum from the early years to build foundational competence.

- Enhance Mentorship Programs: Establish structured mentorship systems pairing medical students with experienced researchers to provide guidance, motivation, and support throughout their research projects.

- Allocate Protected Time for Research: Modify academic schedules to include dedicated, protected time for students to engage in research activities without compromising clinical or academic responsibilities.

- Improve Institutional Support: Develop institutional policies and resources that facilitate student-led research, including streamlined administrative processes, access to research tools, and funding opportunities.

- *Foster a Research Culture:* Promote a positive institutional culture that values research by recognizing student research achievements and encouraging faculty involvement in mentoring and collaborative projects.

### Study limitations

This study has several limitations that warrant consideration. As a narrative review, it synthesizes findings from previously published studies with varying methodologies and quality, which may limit the consistency and comparability of the results. The reliance on studies published between 2010 and 2025 may exclude earlier relevant research, potentially overlooking long-standing barriers or trends. Additionally, the review was restricted to articles published in English, which introduces a language bias and may have excluded valuable data from non-English-speaking regions. The narrative review design, while useful for thematic synthesis, lacks the rigor of systematic reviews or meta-analyses, potentially introducing selection bias and limiting reproducibility. Finally, the conceptual framework developed from the findings, though insightful, has not undergone empirical testing, which restricts its immediate applicability across diverse educational settings.

### Conclusion

This scoping review provides a comprehensive mapping of the barriers that hinder medical students' engagement in research, highlighting how these challenges interact across individual, educational, and institutional levels. In contrast to prior reviews, it synthesizes evidence across multiple regions and study designs, offering a conceptual framework that links identified barriers with actionable strategies, including targeted mentorship, integrated research training, and enhanced institutional support. By presenting this synthesized, policy relevant overview, the review advances understanding of medical student research engagement and provides a foundation for designing interventions that are both context-sensitive and scalable. Future research should focus on evaluating the effectiveness of these strategies in diverse educational settings to strengthen support for emerging clinician-scientists.

## Supporting information

**S1 Appendix. Preferred Reporting Items for Systematic reviews and Meta-Analyses extension for Scoping Reviews (PRISMA-ScR) Checklist.**
(DOCX)

**S2 Appendix. Search strategy.**
(DOCX)

**S3 Appendix. Data extraction table.**
(DOCX)

## Author contributions

**Conceptualization:** L. Winter Mokhwelepa, Tumisang Malete.

**Data curation:** L. Winter Mokhwelepa, Tumisang Malete.

**Formal analysis:** L. Winter Mokhwelepa, G. Olivia Sumbane.

**Investigation:** L. Winter Mokhwelepa.

**Methodology:** L. Winter Mokhwelepa.

**Project administration:** L. Winter Mokhwelepa.

**Resources:** L. Winter Mokhwelepa.

**Software:** L. Winter Mokhwelepa.

**Supervision:** L. Winter Mokhwelepa, G. Olivia Sumbane, Shisana Baloyi, Samuel Risenga.

**Validation:** L. Winter Mokhwelepa.

**Visualization:** L. Winter Mokhwelepa.

**Writing – original draft:** L. Winter Mokhwelepa.

**Writing – review & editing:** L. Winter Mokhwelepa, G. Olivia Sumbane, Tumisang Malete.

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
