## [Decision Letter · Decision Letter 0]

21 Jan 2026

Dear Dr. RATSOMA,

Thank you for submitting your manuscript to PLOS ONE. After careful consideration, we feel that it has merit but does not fully meet PLOS ONE’s publication criteria as it currently stands. Therefore, we invite you to submit a revised version of the manuscript that addresses the points raised during the review process.

We look forward to receiving your revised manuscript.

Kind regards,

Mukhtiar Baig, Ph.D.

Academic Editor

PLOS One

**Journal Requirements:**

1. When submitting your revision, we need you to address these additional requirements. Please ensure that your manuscript meets PLOS ONE's style requirements, including those for file naming. The PLOS ONE style templates can be found at https://journals.plos.org/plosone/s/file?id=wjVg/PLOSOne_formatting_sample_main_body.pdf and https://journals.plos.org/plosone/s/file?id=ba62/PLOSOne_formatting_sample_title_authors_affiliations.pdf 2. Please ensure that you include a title page within your main document. We do appreciate that you have a title page document uploaded as a separate file, however, as per our author guidelines (http://journals.plos.org/plosone/s/submission-guidelines#loc-title-page) we do require this to be part of the manuscript file itself and not uploaded separately. Could you therefore please include the title page into the beginning of your manuscript file itself, listing all authors and affiliations. 3. When completing the data availability statement of the submission form, you indicated that you will make your data available on acceptance. We strongly recommend all authors decide on a data sharing plan before acceptance, as the process can be lengthy and hold up publication timelines. Please note that, though access restrictions are acceptable now, your entire data will need to be made freely accessible if your manuscript is accepted for publication. This policy applies to all data except where public deposition would breach compliance with the protocol approved by your research ethics board. If you are unable to adhere to our open data policy, please kindly revise your statement to explain your reasoning and we will seek the editor's input on an exemption. Please be assured that, once you have provided your new statement, the assessment of your exemption will not hold up the peer review process. 4. We note that your Data Availability Statement is currently as follows: All relevant data are within the manuscript and its Supporting Information files.  Please confirm at this time whether or not your submission contains all raw data required to replicate the results of your study. Authors must share the “minimal data set” for their submission. PLOS defines the minimal data set to consist of the data required to replicate all study findings reported in the article, as well as related metadata and methods (https://journals.plos.org/plosone/s/data-availability#loc-minimal-data-set-definition). For example, authors should submit the following data: - The values behind the means, standard deviations and other measures reported;- The values used to build graphs;- The points extracted from images for analysis. Authors do not need to submit their entire data set if only a portion of the data was used in the reported study. If your submission does not contain these data, please either upload them as Supporting Information files or deposit them to a stable, public repository and provide us with the relevant URLs, DOIs, or accession numbers. For a list of recommended repositories, please see https://journals.plos.org/plosone/s/recommended-repositories. If there are ethical or legal restrictions on sharing a de-identified data set, please explain them in detail (e.g., data contain potentially sensitive information, data are owned by a third-party organization, etc.) and who has imposed them (e.g., an ethics committee). Please also provide contact information for a data access committee, ethics committee, or other institutional body to which data requests may be sent. If data are owned by a third party, please indicate how others may request data access. 5. Please amend either the title on the online submission form (via Edit Submission) or the title in the manuscript so that they are identical. 6. One of the noted authors is a group or consortium. In addition to naming the author group, please list the individual authors and affiliations within this group in the acknowledgments section of your manuscript. Please also indicate clearly a lead author for this group along with a contact email address. 7. Your ethics statement should only appear in the Methods section of your manuscript. If your ethics statement is written in any section besides the Methods, please move it to the Methods section and delete it from any other section. Please ensure that your ethics statement is included in your manuscript, as the ethics statement entered into the online submission form will not be published alongside your manuscript. 8. We note that you have referenced (Rehman MZ, Yahya AW, Imran J, Shehryar M.) which has currently not yet been accepted for publication. Please remove this from your References and amend this to state in the body of your manuscript: (Rehman MZ, Yahya AW, Imran J, Shehryar M. [Submitted]”) as detailed online in our guide for authorshttp://journals.plos.org/plosone/s/submission-guidelines#loc-reference-style 9. We note that Figure 1 in your submission contain map images which may be copyrighted. All PLOS content is published under the Creative Commons Attribution License (CC BY 4.0), which means that the manuscript, images, and Supporting Information files will be freely available online, and any third party is permitted to access, download, copy, distribute, and use these materials in any way, even commercially, with proper attribution. For these reasons, we cannot publish previously copyrighted maps or satellite images created using proprietary data, such as Google software (Google Maps, Street View, and Earth). For more information, see our copyright guidelines: http://journals.plos.org/plosone/s/licenses-and-copyright. We require you to either present written permission from the copyright holder to publish these figures specifically under the CC BY 4.0 license, or remove the figures from your submission: a. You may seek permission from the original copyright holder of Figure 1 to publish the content specifically under the CC BY 4.0 license.   We recommend that you contact the original copyright holder with the Content Permission Form (http://journals.plos.org/plosone/s/file?id=7c09/content-permission-form.pdf) and the following text:“I request permission for the open-access journal PLOS ONE to publish XXX under the Creative Commons Attribution License (CCAL) CC BY 4.0 (http://creativecommons.org/licenses/by/4.0/). Please be aware that this license allows unrestricted use and distribution, even commercially, by third parties. Please reply and provide explicit written permission to publish XXX under a CC BY license and complete the attached form.” Please upload the completed Content Permission Form or other proof of granted permissions as an "Other" file with your submission. In the figure caption of the copyrighted figure, please include the following text: “Reprinted from [ref] under a CC BY license, with permission from [name of publisher], original copyright [original copyright year].” b. If you are unable to obtain permission from the original copyright holder to publish these figures under the CC BY 4.0 license or if the copyright holder’s requirements are incompatible with the CC BY 4.0 license, please either i) remove the figure or ii) supply a replacement figure that complies with the CC BY 4.0 license. Please check copyright information on all replacement figures and update the figure caption with source information. If applicable, please specify in the figure caption text when a figure is similar but not identical to the original image and is therefore for illustrative purposes only.The following resources for replacing copyrighted map figures may be helpful: USGS National Map Viewer (public domain): http://viewer.nationalmap.gov/viewer/The Gateway to Astronaut Photography of Earth (public domain): http://eol.jsc.nasa.gov/sseop/clickmap/Maps at the CIA (public domain): https://www.cia.gov/library/publications/the-world-factbook/index.html and https://www.cia.gov/library/publications/cia-maps-publications/index.htmlNASA Earth Observatory (public domain): http://earthobservatory.nasa.gov/Landsat:
http://landsat.visibleearth.nasa.gov/USGS EROS (Earth Resources Observatory and Science (EROS) Center) (public domain): http://eros.usgs.gov/#Natural Earth (public domain): http://www.naturalearthdata.com/ 10. If the reviewer comments include a recommendation to cite specific previously published works, please review and evaluate these publications to determine whether they are relevant and should be cited. There is no requirement to cite these works unless the editor has indicated otherwise. 

Reviewers' comments:

**Comments to the Author**

1. Is the manuscript technically sound, and do the data support the conclusions?

Reviewer #1: Partly

Reviewer #2: Yes

2. Has the statistical analysis been performed appropriately and rigorously?

Reviewer #1: No

Reviewer #2: N/A

3. Have the authors made all data underlying the findings in their manuscript fully available?

Reviewer #1: No

Reviewer #2: No

4. Is the manuscript presented in an intelligible fashion and written in standard English?

Reviewer #1: Yes

Reviewer #2: Yes

**Reviewer #1:** The manuscript contains multiple internal numerical contradictions that undermine the credibility of the review process. The PRISMA flow diagram reports 400 records assessed for eligibility and 286 excluded, which mathematically should result in 114 eligible studies; however, only 57 studies are reported as included, with no explanation for the remaining 57 records. This represents a fundamental breakdown in PRISMA transparency.

There are inconsistent study counts across analytic sections. While the manuscript repeatedly claims inclusion of 57 studies, the geographical distribution presented in the Results section accounts for only 49 studies, leaving eight studies unaccounted for. Similarly, the study design breakdown lists 49 cross-sectional, 4 mixed-methods, and 2 qualitative studies, totaling 55 rather than 57, with no clarification regarding the missing studies.

The exclusion criteria are contradictory across sections. The Methods section states that studies published prior to 2010 were excluded, whereas the PRISMA diagram reports exclusion of studies published prior to 2000. This discrepancy renders the screening framework factually inconsistent and unreliable.

There is a temporal impossibility in the citation corpus. The Methods indicate that the literature search concluded in March 2025, yet the manuscript cites multiple studies published in April and May 2025. The inclusion of post-search publications contradicts the stated search protocol and compromises methodological integrity.

The manuscript exhibits reference indexing and citation mismatches. Appendix B assigns incorrect citation numbers to multiple studies, including misalignment between in-text citations and the reference list. In addition, at least one reference (Al Absi et al.) appears duplicated under different citation numbers, artificially inflating the reference count.

There are serious concerns regarding citation authenticity. At least one cited author (“Kingpriest PT”) appears anomalous and cannot be readily verified in standard biomedical databases, raising concerns about possible fabricated or hallucinated references. Several citations also contain implausibly precise future publication metadata, suggesting predictive placeholders rather than verifiable publications.

The PRISMA diagram reports an excessively high full-text retrieval failure rate, with 90 of 400 eligibility-stage records (22.5%) reportedly inaccessible. This level of retrieval failure undermines claims of a comprehensive review and raises questions about institutional access and search rigor.

The manuscript suffers from methodological inconsistency in review type. It alternately describes itself as a scoping review and a narrative review, despite these being distinct methodological approaches with different standards, further confusing the analytic framework.

There are numerous language, typographical, and template errors throughout the manuscript, including incorrect words, unintelligible phrases, misspellings, and unreplaced placeholder text (e.g., “XXX”), which detract from scholarly professionalism.

The visual materials are of inadequate quality for scientific publication. One figure is a low-resolution map explicitly labeled “Powered by Bing,” and another is a generic schematic that adds no analytical content beyond restating textual headings.

Collectively, these numerical inconsistencies, temporal contradictions, citation irregularities, and reporting errors represent systemic flaws rather than isolated oversights, substantially compromising the reliability and reproducibility of the review.

Your scoping review identifies 'time constraints,' 'lack of mentorship,' and 'institutional barriers' as key themes. This recently published qualitative study (2025) investigates these exact barriers within a specific medical education context (Pakistan). https://doi.org/10.1186/s12909-025-07185-9 Including this citation would provide up-to-date empirical support for your 'Results' section and allow you to triangulate your scoping review findings with recent qualitative evidence regarding the 'dual-focus nature' of medical training that you discuss."

In your 'Study Recommendations' and 'Discussion' sections, you emphasize the need to 'Integrate Research Training Early' and improve research skills. https://doi.org/10.1155/2024/1271802 This study offers a specific pedagogical intervention (peer feedback on research writing) that addresses the 'inadequate training' barrier you identified. Citing this would strengthen your argument for practical, low-resource interventions that institutions can adopt to support student research writing and skill acquisition."

**Reviewer #2:**  Reviewer comments PONE

Thank you for the opportunity to review and learn from this manuscript, The Problem Is Escalating: Barriers Faced by Medical Students in Conducting Research. The authors tackle a topic of clear relevance to medical education globally, particularly at a time when research literacy and scholarly engagement are increasingly viewed as core competencies for future physicians. The manuscript is positioned as a scoping literature review following JBI guidance and aims to synthesise the barriers that limit medical students’ engagement in research, with the stated intention of informing educational reform and policy interventions. The topic is well aligned with the journal’s scope and has potential value for educators, curriculum designers, and institutional leaders seeking to strengthen undergraduate research ecosystems.

Overall, the manuscript addresses an important and widely recognised issue within medical education. The use of a scoping review approach is appropriate given the breadth of the topic, and the authors demonstrate awareness of established methodological frameworks (JBI). The identification of common barriers such as limited mentorship, academic workload, and insufficient research training will resonate with many readers and aligns with experiences reported across diverse contexts. However, some areas require further clarification and revisions as necessary. See bellow:

1. While the introduction effectively establishes the importance of research in medical education, the framing remains largely descriptive. The manuscript would benefit from a clearer articulation of: why a scoping review is particularly warranted at this point in time, and how this review advances understanding beyond existing narrative or systematic reviews on student research engagement. Consider sharpening the rationale to explicitly state the gap this review addresses (e.g., fragmentation of evidence, lack of synthesis across regions, or absence of policy-relevant mapping).

2. The authors lists databases and keywords used; however, the description remains high-level. Boolean operators, truncation, and database-specific adaptations are not described.

Google Scholar is included, but the approach to screening (e.g., number of results reviewed) is not specified. I personally find Google Scholar highly unreproducible for review studies. I would advise you consider providing a more detailed search strategy (either within the manuscript or as a supplementary file), sufficient to allow replication.

3. The review restricts inclusion to English-language studies published between 2010 and 2025. While this is reasonable, the implications of this choice are not discussed on may shape the scope of findings.

4. It is unclear how many reviewers conducted screening and whether screening was performed independently or in duplicate. Given the interpretive nature of categorising “barriers,” this detail is important. Clarify the screening process (number of reviewers, independence of screening, and how disagreements were resolved) to enhance methodological rigor and transparency.

5. The results section identifies several key barriers; however, the analytical process underpinning their identification is not fully elaborated. It is unclear whether thematic synthesis was inductive, deductive, or mixed. The relative weight or frequency of barriers across studies is not described. I think it would be helpful.

6. The discussion appropriately connects findings to educational reform and institutional responsibility. However, some conclusions appear broader than what the scoping evidence alone can substantiate. consider moderating claims related to policy reform by clearly distinguishing between findings derived from the review and interpretive implications offered by the authors.

Minor comments

Minor grammatical and formatting issues are present and would benefit from careful copy-editing.

The abstract could more clearly indicate that this is a scoping review rather than a systematic or effectiveness review.

The conclusion may be strengthened by briefly restating the unique contribution of the review rather than reiterating general challenges already described.

**Do you want your identity to be public for this peer review?** For information about this choice, including consent withdrawal, please see our Privacy Policy

Reviewer #1: No

Reviewer #2: **Yes:** Udoka Okpalauwaekwe

---

## [Author Response · Author response to Decision Letter 1]

26 Jan 2026

Reviewer response document is attached.

---

## [Editor Report · Decision Letter 1]

4 Feb 2026

The Problem Is Escalating: Barriers Faced by Medical Students in Conducting Research; A Scoping Literature Review

PONE-D-25-58067R1

Dear Dr. Mokhwelepa,

We’re pleased to inform you that your manuscript has been judged scientifically suitable for publication and will be formally accepted for publication once it meets all outstanding technical requirements.

Kind regards,

Mukhtiar Baig, Ph.D.

Academic Editor

PLOS One

---

## [Editor Report · Acceptance letter]

PONE-D-25-58067R1

PLOS One

Dear Dr. Mokhwelepa,

I'm pleased to inform you that your manuscript has been deemed suitable for publication in PLOS One. Congratulations! Your manuscript is now being handed over to our production team.

Kind regards,

on behalf of

Professor Mukhtiar Baig

Academic Editor

PLOS One